# Leaving No One behind in Healthy Ageing: A Unique Sub-Group, the “Cardboard Grannies of Hong Kong”

**DOI:** 10.3390/ijerph19159691

**Published:** 2022-08-06

**Authors:** Crystal Kwan, Ho-Chung Tam

**Affiliations:** Department of Applied Social Sciences, The Hong Kong Polytechnic University, Hong Kong, China

**Keywords:** informal waste picking, recycling, cardboard grannies, healthy ageing, Hong Kong, qualitative

## Abstract

The older adult population in Hong Kong is large and diverse. The “Cardboard Grannies” in Hong Kong are informal waste pickers (IWPs) who represent a unique sub-group that is often forgotten in society. This group has unique social and economic conditions and contributions that are not monitored and recognized. Leaving no one behind in healthy ageing requires an understanding of the needs and contributions of those on the margins of society, like older adult IWPs. This study answers two main research questions: (i) what are the service needs of the older adult IWPs and (ii) what are their contributions (social impact)? Qualitative methods were used to collect data from the older adult IWPs and key informants. Thematic analysis and word clouds were used to analyse the data. Nine themes were identified, providing relevant and significant insight into the service needs of the older adult IWPs. Two themes were identified, providing insight into the contributions (social impact) of the older adult IWPs. These themes inform recommendations that cover a range of individual, family, and community service responses to address healthy ageing of this unique sub-group.

## 1. Introduction

### 1.1. Leaving No One behind in Healthy Ageing

From 2015 to 2050, the 60+ population will nearly double and reach 22% worldwide [1]. In Hong Kong, by 2036, the 65+ population will amount to 31% (2.37 million) [2]. Given this demographic shift, policy- and decision-makers worldwide and locally have recognized the need to be proactive in addressing the various needs of the ageing population. The World Health Organization (WHO) [3] declared that 2020–2030 will be the “Decade of Healthy Ageing”. Furthermore, the WHO aligns their strategy with the Sustainable Development Goals (SDGs) that pledge that “no one will be left behind and that every human being will have the opportunity to fulfil their potential in dignity and equality” (p. 1). Locally, in Hong Kong, healthy ageing was a policy direction adopted in 2001 by the Elderly Commission [4] and continues to be a strong focal area. 

Healthy ageing is broadly defined as optimizing both physical and psycho-social well-being and seeing aging as both a process of needs and opportunities [3]. While this concept was adopted from the western context, the GovHK recognized in their initial healthy ageing policy campaign that “efforts to promote Healthy Ageing in Hong Kong should be sustained and perhaps further refined as we gain more experience” [5] (para. 11). Thus, acknowledging the need to localize and contextualize healthy ageing to the sociocultural and diverse population of Hong Kong, this study is a response to the global and local policy frameworks in generating data and knowledge and seeking to inform innovative local responses to healthy ageing for a forgotten and marginalized group of older persons—the Cardboard Grannies of Hong Kong.

### 1.2. Who Are the “Cardboard Grannies” of Hong Kong? 

The “Cardboard Grannies” of Hong Kong are older adult informal waste pickers (IWPs) whose social and economic condition and contributions are not monitored and recognized. There is stigma attached to people who take on this type of work as their livelihood. However, they can be perceived as “service providers in sociotechnical systems (collecting and recycling urban waste), and as economic actors (critical to the value chain)” [6] (p. 376). Most of the literature on IWPs focuses on cities in low and middle-income countries (e.g., Brazil, Columbia, South Africa, Chile, and India), who have a long history of organizing such workers to advocate for greater rights and protections and whose public/municipal waste management systems are still developing [6,7]. 

Hong Kong is a developed city within an upper-middle income country (China) that has a developed a public municipal waste management system. However, the phenomenon of IWPs persists. Furthermore, there is a distinct age and gender dimension to the IWPs in Hong Kong. A study conducted by the Waste Pickers Platform (WPP) in March 2018 highlighted that of the 505 waste pickers surveyed in 11 of the districts in Hong Kong, 80% were women and 82% were 60 years of age and older [8]. Thus, locally, the informal waste pickers are known as the “Cardboard Grannies”. 

The WPP found that of those surveyed, the average monthly income from selling the recyclable scraps was HK$716. Twenty five percent of the interviewees relied on such income to meet their basic needs, and more than 60% stated that they made less than HK$5000/month when combining such income with others (e.g., wages from other employment/livelihoods and/or government social welfare). Furthermore, 90% of the respondents identified that economic reasons drove their decision to engage in such livelihoods. Aside from this one study conducted by an advocacy group, there is minimal studies in the grey literature and no known studies in the peer-reviewed literature focusing on this marginalized group of older persons. This study aims to address this knowledge gap.

### 1.3. Research Objective and Questions

The objective of this study is to develop a practice framework that informs healthy ageing interventions for a forgotten and marginalized group: the older adult IWPs of Hong Kong. This study answers two primary research questions: (1) what are the service needs of the older adult IWPs of Hong Kong and (2) what are their contributions (social impact)? 

## 2. Materials and Methods

### 2.1. Sample, Recruitment and Data Collection

This study was approved by the Human Subjects Ethics Committee at the Hong Kong Polytechnic University (Reference Number: HSEARS20210806005). Informed consent was obtained from all participants of the study. This was a qualitative study engaging two perspectives: older adult IWPs (*n* = 26) and key informants, who interact with the older adult IWPs (e.g., staff from the recycling depots, shop owners in the neighbourhoods, district councillors, service providers/social workers, former staff of the Food and Environmental Hygiene Department (FEHD) (*n* = 15). A local non-governmental organization (NGO) providing services to older adult IWPs in Hong Kong helped us to recruit participants and referred us to key informants as well. The total sample size was *N* = 41. Two data collection methods were used: semi-structured individual interviews and focus groups. The individual interviews were used to collect data from 20 older adult IWPs, and 15 key informants. Each interview was approximately 1 h long. The focus group discussion was used to collect data from 6 older adult IWPs and lasted approximately 1.5 h. The interviews were conducted at a location chosen by the participants, and the focus group was conducted at a local senior centre. The interviews and focus groups were audio-recorded and then transcribed verbatim. All data was collected from September 2021 to April 2022. 

### 2.2. Data Analysis 

Thematic Analysis (TA) [9] was used to analyse the data. A constant comparison method [10] was used in coding. The first author created a codebook based on the first five transcripts of each perspective (older adult IWP and key informant) and conducted the majority of the coding. The second author used the codebook to assist with some of the subsequent coding. Any new emergent codes created were discussed by the two analysts and the codebook was updated accordingly. NVivo (QSR International (UK) Limited, London, UK) was used to systematically organise and analyse the large amount of data. Additionally, word clouds, which provide a visual representation of the frequency of words used in a set of text [11], were used to analyse two specific interview questions that were asked in a finish-the-sentence format: (i) One word or phrase to describe older adult IWPs (asked to key informants) and (ii) Without older adult IWPs, Hong Kong would be (asked to the older adult IWPs) …

## 3. Results

### 3.1. Participant Characteristics (N = 41)

The older adult IWPs totalled *n* = 26. The majority of the older adult IWPs (*n* = 21) are female, and mainly from the middle-old (*n* = 14) and old-old (*n* = 8) category. Most of them are from China (Mainland) (*n* = 23), yet have lived in Hong Kong for a decade or more (*n* = 21). Eighteen of them live in public housing, and all participants who are living alone are female (*n* = 13). Of those who did identify their family status, only two of them do not have children and 12 are widowed. More than half of them have been picking for 10+ years (*n* = 14). See Appendix A for details of the older adult IWPs.

The key informants totalled *n* = 15. Of the 15 key informants, 5 of them are staff and social workers from community-based/non-governmental organisations. Three of them are from the recycling industry, which include owners of recycling depots and board of directors in the industry. Other key informants include District Councillors, former FEHD officers, labour artists, researchers and shop owners who frequently interact with the older adult IWPs. See Appendix A for details of the key informants. 

### 3.2. Service Needs: Perspectives from Older Adult IWPs and Key Informants

Nine themes were identified from the qualitative data of the older adult IWPs and key informant interviews providing relevant and significant insight into service needs. 

#### 3.2.1. Theme 1: Social Relations with Others at the Worksite/Neighbourhood 

A major theme identified across all older adult IWPs (*n* = 26, 448 references) and almost all key informants (*n* = 14, 135 references) and the most discussed topic in the interviews relates to the social relations with others at the worksite/neighbourhood. Both positive and negative experiences with others (e.g., FEHD officers, cleaners, other pickers, staff of recycling depot, shop owners, and the public) were shared. However, more negative experiences were shared than positive. The study participants shared how such interactions and relationships implicated the older adult IWPs’ work and well-being in various ways. 

##### Positive Experiences 

When it comes to sharing positive experiences and interactions, many of the older adult IWPs talked about how others would go out of their way to make it more convenient for the older adults to pick. As one participant shared, “they just take the cardboards to me. To pick is quite tough, running around. So, they take it to me. It’s just a neighbourhood business, the hardware shop which has cardboards left and they bring it to me.” (I3). 

Other older adult IWPs shared how there are known in their neighbourhood and how others helped them to make their job more convenient. As one participant said, “One policeman is so nice too. He often brings some cardboards to me and puts them outside my flat.” (I4). A 93-year-old female IWP confessed that she cannot miss a day at her worksite (collecting recyclables at the MTR station) because of the strong social connections and relations she has with the commuters. She said, “My friends miss me if I don’t go there for a day! Not just one or two people, but many people miss me! Yes, they are like my children ha ha.” (I13). She further shared a story about a mother and daughter who commuted regularly at the MTR station and whom she has known for many years,

“When she sees me she always asks her mom to help me. Her mom always helps me to push my trolley when she comes back. Yes, she cares about me a lot. It’s so rare to meet nice people like her. She helps me to push the trolley to the entrance. They can find me in MTR station. Yesterday or last Friday, she found me and told me she was studying university. She knew me since kindergarten. I knew her since kindergarten!” (I13).

The key informant interviews also revealed both positive and negative interactions at the worksite for the older adult IWPs. Regarding the positive interactions, a key informant from the recycling industry revealed that he would actually pay more to the older adult IWPs when they come to his depot,

“I don’t know about other people. When the grannies come, we pay them accurately by weight. The other place might have different pay rates according to how good they deem the quality of the cardboard. If the quality isn’t great they might pay less. But we don’t do that. Sometimes, if we see the grannies come I will pay them a bit more.” (I2).

Another key informant (shop owner) shared how he feels about the older adult IWPs, “I fully support their work. Even in front of my shop or the grocery shops nearby in the market.” (I04). Another shop owner shared how some neighbourhood shops would even provide the trolleys for the older adult IWPs to use, “Actually, there are some shops providing trolleys to waste-pickers to work. You know buying a trolley costs a lot for some waste-pickers.” (I05).

##### Negative Experiences

The negative experiences and stories shared by both older adult IWPs and key informants relate to three interrelated subthemes: (1) insecure working spaces, (2) conflict with FEHD and other pickers, and (3) stigma attached to picking.

Insecure Working Spaces

Both key informants and older adult IWPs shared how a large component of the older adult IWPs’ work is related to sorting the recyclables, deconstructing the cardboards, and packing the cardboards on the trolleys. As one key informant from the recycling industry shared,

“Waste-picking is time consuming. Cutting and packing cardboards are time consuming. For example, the cardboard boxes disposed by the pharmacy, which also have other waste [non-recyclables] in the boxes have to be categorised, cut and packed. The time cost is high.” (I07).

This time-consuming process leads to the need for a secure and sufficient space in the neighbourhood for the older adult IWPs to work. However, the older adult IWPs shared a lot of issues regarding finding a secure space to do their work and to store their recyclables safely. One older adult IWP shared how she does this processing in an alleyway and will sleep there to guard her trolley full of recyclables, 

“My things are in the alley so I can pick and organize. I have no time. Yeah, I have to do it every day. Little by little. I need to empty them by Lunar New Year, so I can take my trolley away. Yeah, I sleep here to watch my trolley. I worry about my trolley being stolen. That’s all. So I sleep on the street, which is quite comfortable.” (I14).

Another participant shared that she also sleeps on the street sometimes to guard her recyclables,

“You know I could still put the recyclables at home when my children were still small. They are now married and some of them even have kids. So I can’t put the recyclables at home, and I have no choice but to put the cardboards on the streets. I waste pick a lot. I usually sell the cardboards in the early morning and then I continue picking. After that, I go to get free lunchboxes and have lunch outside. After that, I continue picking until late night when everyone finishes working. After picking, then I have to pack and sort the cardboards. You know, sometimes I sleep with the cardboards on the street!” (P5 FGD).

Some of the key informants suggested there is a need for consumers or shop owners to do the extra step of pre-sorting the recyclables and deconstructing the boxes to make it easier for the older adult IWPs. For instance, a key informant from an NGO suggested that there needed to be more public education about taking responsibility for being part of the recycling system, 

“Unlike Taiwan, there is no education about the ecology and recycling in Hong Kong. In other sense, everyone is a waste-picker because everyone has the responsibility to do recycling. This is the healthiest development of recycling. But in Hong Kong, only the older people are doing recycling.” (I06).

The issue of working spaces was discussed a lot by the key informants. One district councillor opined “You have to tidy up the area as well to consolidate the boxes, and where would you put the recyclables? Where to put it? That is the biggest question.” (I08).

Several key informants shared that a possible solution is to have designated public spaces for the older adult IWPs to do their work, but at the same time they recognized the limitations to this solution. As one district councillor opined, 

“The most effective thing we can help is to provide an area in public spaces for them but there are so many drawbacks. It depends on public spaces which are limited, you know. Sometimes the cardboards are organised but sometimes it could be very messy. Of course it is not that easy because there are opposing voices from other co-workers and it involves different government departments such as the Lands Department. Also, the departments are afraid of taking the responsibility when there is an accident.” (I11). 

Interestingly, when an older adult IWP was asked about creating a zone for pickers to process and do their work, the participant shared “but we will be embarrassed.” (I09), highlighting another challenge to this solution. 

2.Conflict with FEHD and Other Informal Waste Pickers

Almost all of the older adult IWPs shared their ongoing struggles with the FEHD officers. One participant shared how one of his biggest loads was confiscated from them,

“Last year, I collected a lot of stuff during the New Year. And then people from the Department [FEHD] threw them all away. Yeah. They took it all away. That year they took their holiday earlier. The authorities took them all away.” (I12).

The confiscation of the older adult IWPs’ trolleys (which helps them to collect more and transfer their recyclables across distances easier) are an ongoing problem for the participants. One participant shared it was her greatest worry even over her health, 

“That time when they took three of my carts at once, I called the bosses or like manager at the waste collection station and asked them if they had seen my cart. I was like how did they have the nerve to do that? They took away my properties and confiscated them. Do you give me money to buy my stuff back? I worked so hard to earn and save that money to buy the cart. They took so many carts. It’s a couple thousands (of HK$) and are they going to give me back that money? Are you even human? There was nothing on the carts. I put locks on them. No one was going to walk there and I’m not in the way of something, but they still took my things away. It’s not like they paid me for my cart. I paid for it myself, right? You cannot just do that! I cannot even get it back. What do I do? Of course, this is the most worrying. The second thing I’m most worried about is my health.” (I15).

However, there was one participant who shared a unique and positive relation with the FEHD, 

“Those staff [from the FEHD] said there was nothing wrong for me to scavenge cardboards. They said I could earn money and I helped the environment! They come because they have to fulfil their duty. They are so nice to me. When those staff come, they were so polite and they asked me to remove things for them to take a photo, just for their duty. Yes, they are so nice to me every time.” (I10).

Aside from the ongoing conflicts and struggles with the FEHD, the older adult IWPs also shared negative interactions with other IWPs. One participant shared the following about a group of other IWPs in his neighbourhood, 

“Yeah, they [other pickers] say that it’s theirs. What they are saying is basically that it is their jurisdiction. That I am not to intrude in their area. But I’m like, I don’t go and take your stuff. These are all public places. When I see them picking somewhere I would not go near. But they always occupy the spaces, making it impossible for me to collect near them. And they come near me when they see me collecting things.” (I12).

Another shared the following,

“Sometimes I have a deal with the shop keepers who would save the cardboards for me. But some staff there know other pickers and they would give it to them instead. I was angry when I knew that. I told the shop keeper, and she promised to keep an eye on it for me. The staff was upset about me because she could not give it to the person from her village. The other picker argues with me every day. It is about her greediness. She only takes thick cardboards and throws away thin cardboards. She parks the trolley to my place and blocks my way.” (I14).

A key informant (shop owner) shared the following regarding the territorial rules for picking, “They have their own areas. If you throw the cardboards in the rubbish bins, everyone can pick them. But usually each waste-picker has their own territory.” (I05).

3.Stigma Attached to Picking

Many of the older adult IWPs shared that they are not concerned with the negative image related to their work, but they are well aware that others may feel this way towards them. As this participant shared, “They think I am low-class because I scavenge cardboards. I don’t care! I don’t care! They don’t talk to me because I scavenge cardboards. It’s okay, never mind them.” (I02).

Another participant echoed this sentiment,

“We are at this age where we don’t mind how they see us. Whether you think I am bad or good I will just mind my own business. It’s all about earning to me. I don’t care about what they think. I don’t disgrace myself even if people look down upon me. That’s all.” (I03).

A key informant from the recycling industry suggested that the reason why there are more female pickers is due to the stigma attached to the work,

“Generally, there’s more women. Well, men have to save face don’t they? Some of them at least. I heard from one lady that the other day she asked her husband to bring some cardboard over. Her husband works at the fruit market and he doesn’t deliver to the door. He was scared of people seeing him here.” (I02).

Some of the key informants shared that addressing the stigma attached to this work is one way of supporting this group. A district councillor shared the following when asked what is the most important support that the pickers need currently, “Recognition. Yeah, it’s important. Of course it has to be done step by step. But everything stems from stigma. If you don’t give them recognition, a lot of problems arise.” (I08).

#### 3.2.2. Theme 2: Picking Provides a Trifecta Experience 

Another key theme, identified across 26 older adult IWPs (316 references) and 12 key informants (58 references), relates to the reasons why the older adult IWPs pick as their key activity. There were three main reasons why they pick as a primary activity: (1) for meaning/purpose, (2) for income, and (3) for exercise. 

##### For Meaning/Purpose

From the older adult IWPs perspective, picking was a meaning-making/purposeful opportunity. Of the three reasons, this one was discussed the most (*n* = 20, 137 of references). One older adult IWP shared “I wake up and I have something to do, it is the happiest.” (I10). Another older adult IWP explained,

“Well, they [her family] have to go to work, so I’m home alone most of the time. So I have a lot of things to think about by myself. If they are back then I have people to talk to and I can chat. If not, then there’s nothing to do. There’s nothing to do other than facing the four walls so what’s the point then.” (I15).

Another older adult IWP echoed this sentiment, 

“If I don’t come here [referring to the neighbourhood in which she picks] I’m just idle at home with nothing to do. It’s not about the money. I don’t care about that. I just want something to do. When I am falling asleep and thinking about the day, and I know that I have done something for the day, I feel happy. I feel energized.” (I16).

This sentiment was also shared by many key informants, as this shop owner opined, 

“From my observation actually they don’t need the income from waste-picking to sustain their livelihood, but they still do it every day. To be fair, the government provides living allowances for them, but they don’t care about this. Take an example of this granny, she could barely walk because her legs have issues. She said it takes 30 min for her to walk back home. She still does waste-picking every day, but if you ask her if she is really financially insufficient the answer is no. People have different goals in life probably.” (I04).

##### For Income 

A number of older adult IWPs (*n* = 19, 118 references) also shared how picking enabled them to have a source of income, and this reason was also discussed a lot. Many of them shared they had full-time jobs before but were let go due to their old age. They said they turned to picking because no one else would hire them. An older adult IWP shared, 

“I worked in [Name of Company] Estate Agency Limited. I needed to retire when I was 65 years old. They didn’t renew my contract. I needed to still support my family, so I scavenged cardboard. No one would hire me you know even if I want to clean toilets! Of course, I would choose a full-time job. You can gain at least HK$10,000 a month, right? Minimum wage HK$30–40, right? I can only earn around HK$30 a day. Yes, if I could choose I would find a job. But no one will hire me.” (I05).

Another older adult IWP echoed this sentiment and shared how he did actually look for other jobs but was unsuccessful, “I have tried to seek for jobs. They asked about my age. I told them I am more than 70 years old. So they rejected me.” (I07).

The amount of income earned through picking is minimal and would only be supplementary to other sources of major income, which were often government financial social welfare (e.g., Old Age Living Allowance [OALA], Old Age Allowance (OAA), Comprehensive Social Security Assistance [CSSA], and Disability Allowance [DSA]). The income from picking would vary greatly ranging from HK$10 to $200/day, with most of the pickers in the lower range (HK$10–$100). One older adult IWP shared she earned “a few hundred [HK$] in a month only. Yes, extra money for food ha ha.” (I06). Another older adult IWP echoed this sentiment, “I can use this money to buy food. It can help my family. They don’t need to buy food for me.” (I04). Another participant shared how the income earned is only enough for vegetables, “I can buy a few catties [a traditional unit used in Hong Kong equivalent to about 600 g] of vegetables. But I can’t buy one catty of fish! I can’t buy one catty of pork too. HK$68 for one catty of pork. I can’t afford that, right?” (I07). 

Notably, for the older adult IWPs who do not have public housing and do not receive government financial assistance, income was the main reason for picking. The financial challenges are illustrated through this participant, who applied and is waiting to get into public housing. She shared, 

“I would like to stop scavenging cardboards, but how can I live? No choice. I have to pay for rent. HK$4000 is difficult for me as I don’t have a job. I use all my money for rent and nothing is left. How can I maintain my daily life and [get] meals!? No I can’t. It’s so annoying. The most important thing for me is to get public housing now. If I can get public housing I don’t have to pick. I am still waiting for it. I have applied for the normal and express flat allocation. I don’t know when I will get it though.” (I02).

Picking for income was also shared by the key informants (*n* = 10, and 36 references) as a reason behind doing this informal work. One key informant from the recycling industry explained, 

“You know Hong Kong has the CSSA scheme (social assistance) and people who receive CSSA cannot go to work (because they will be ineligible to receive CSSA once they have certain amount of income). Therefore they choose to do waste-picking to earn more income as the money from CSSA is not enough for their livelihood.” (I07).

##### For Exercise

Exercise was another key reason for picking from the older adult IWPs’ perspective (*n* = 15, 57 references). A participant shared the income earned was too low to be the main driver of picking, “[picking provides a] very low income, but I take it as exercise. Nothing else.” (I08). Another participant echoed this sentiment, 

“Mainly for exercise. I bring my trolley and walk around, no matter if there are items or not [to pick]. I feel better as I walk more. My knees used to give me pain. Yes, no pain now. If I don’t walk I feel painful.” (I06).

Another participant shared that he would often get stopped by strangers while doing work and they would offer him money as a kind gesture but he would reject the offer and explained, “Yes many people walk around and they donate money to me. They give HK$100 for me to Yum Cha [refers to the social activity of drinking tea and eating dim sum], But I reject and I say thank you to them. I treat scavenging cardboards as an exercise.” (I10).

There were a few key informants (*n* = 4, 8 references) who also shared this perspective of picking for exercise. As a social worker serving the participants shared,

“We find out more reasons why the older people are doing waste-picking. The other main reason (aside from income) is that it is good for the body and it is like a kind of exercise. It makes you healthier let’s say mentally. So some of the older people insist to work because it makes them healthier and happier and they say they can sleep better.” (I09).

##### Trifecta Experience

Meaning/purpose, income, and exercise were three key reasons why the older IWPs continued to pick. Importantly, many of the older adult IWPs shared that picking enabled all three goals (meaning/purpose, income, and exercise) to be achieved simultaneously, creating a trifecta experience. As an older adult IWP shared, “Most importantly, I can do some physical exercise, and have something to do. I have income and I can help others! Right?” (I03). Another participant echoed this, “I can kill time, have a little income and just take a walk. I can walk and scavenging cardboard can be a kind of exercise”. (I04). Another participant shared “I do this as an exercise only. You can’t earn a lot. But I feel happy that I can earn a little from it. I can meet my friends during work. If I don’t come for one day then they ask why I didn’t.” (I13). 

The need for the trifecta experience is further illustrated by older adult IWPs who shared that there was no shortage of activities they could do at senior centres, but they prefer to pick. An older adult IWP explained, “Yes I can [join the activities at the centre] but I can’t go if I scavenge cardboard. They [service providers] invite me to join. Yes, making dumplings, watering plants, I can join these. Someone invites me to join every week! I don’t join.” (I11). Another participant shared how even if a free trip was offered to him he would refuse, “Even if you invite me for a free trip I will not go! Because if someone picks those items I can’t get them back.” (I05).

#### 3.2.3. Theme 3: Physical Decline and Medical Health Needs 

Another key theme identified across 22 older adult IWPs (146 references) and 13 key informants (25 references) relates to the physical decline and medical health needs of the older adult IWPs. Many of them shared how they are experiencing a variety of health issues (e.g., heart problems, headaches, eye problems, allergies, bodily pains, muscle pain, leg pain) and some shared how they were on numerous medications. One older adult IWP illustrated, “I take medication in the morning and at night. Nine pills in the morning. Three pills at night. Yes ha ha ha 9 pills!” (I6). Another participant shared a similar situation, 

“Medicine every day. Every day I take a lot of medications. They told me that I have to get injections. I don’t want to do that. If you started injections you can’t just stop. You must maintain it and keep injecting. It’s too troublesome to me. That’s why I’m on another kind of drug. My dosage has been increasing. I just need to take one pill each time in the past and now I have to take two of them, ha ha ha, and I have four different kinds of medications to take. Yeah, morning and night.” (I12).

Another participant shared how she has been on medication for over a decade, 

“The doctor said I have “three highs” -high blood pressure, high fat [cholesterol] and high blood sugar [diabetes]. The doctor prescribed medication for me to control it. He said I needed to take it, otherwise I will get into trouble. So I’ve been on medication for more than 10 years, but I have not increased the dose.” (I11).

Another older adult IWP complained about dermatological issues, saying “[I have] itchy skin, and it has lasted for 20 years. I can’t manage it.” (I04).

Aside from existing health issues, many of the older adult IWPs shared how the stress and physical demand of picking is taking a toll on their aging bodies. As one participant said, “You need to bundle them and other procedures. Scavenging cardboard is so hard. I was exhausted during the procedures. I need to tear up, cut, and do other procedures, like folding them well. I sweat a lot during scavenging!” (I04). Another older adult IWP echoed this sentiment, 

“Scavenging cardboard is hard because you need to bend over to pick cardboard. If there are many cardboards I need to grip a chair. I get a painful wrist and backbones. I feel painful if I walk, but I don’t have a solution. My legs and wrist feel aching during walking.” (I06).

Another participant opined that picking was harder than when she used to farm, as she explained, “Yes, it looks like it’s easy. But things are heavy. It hurts your back. It’s even worse than being a farmer back home. I am always curling up my back.” (I15).

A key informant district councillor also shared his concerns regarding the older adult IWPs’ old age and the physically taxing and dangerous circumstances of their job. He said,

“They actually can’t sustain their work. They’ve told me this. Even the younger ones always say that their bodies have problems. I know that they get scratches and sprains easily. It’s so dirty. It’s different for them if they get scratched. Their immune systems aren’t great. If they get inflammation then they need at least a couple weeks to recover. It’s totally different from when we get hurt.” (I08).

Most of the key informants’ discussion was related to the occupational health hazards of picking. As one key informant from the recycling industry shared, 

“I usually tell waste-pickers to be aware of the traffic. The most dangerous thing for waste-pickers is that they usually walk between the gaps of cars. Most older people are relatively short and the truck drivers can barely see them. So I always tell them to walk in front of the cars instead of behind and at the sides so the drivers could see them. The cars could hit them easily.” (I07).

He further shared that they even gave some reflective vests and hats for the older adult IWPs to enhance their visibility on the streets. He explained, 

“We actually gave them reflective vests but many of them refuse to wear them. So sometimes we give them hats where the edges are reflective. They are happier to wear the hats because it is hot in the summer and the hats can protect them from the sunlight and heat.” (I07).

Providing protective gear and education was echoed by another key informant (an NGO worker). Albeit he also noted the challenges with this idea, 

“In general, I think we can provide them with gears such as gloves, cutters, reflective vests, boots, trolleys, etc. Also, we can also provide them with education of work safety. But the point is whether they would accept these supports.” (I09).

The older adult IWPs shared mixed feelings about wearing protective gloves. Those who did not wear gloves provided reasons such as “I don’t wear gloves because wearing gloves makes you so clumsy;” (FG P4). “It is more flexible not wearing gloves while working, like you can do whatever you want;” (FG P3) and “I don’t [wear gloves], it makes me less productive.” (FG P2). There were others who shared they would wear the gloves at certain times of the picking process as one participant said, 

“Yes, but it [gloves] comes in the way when I try to tie stuff. But I do wear it when I breakdown the cardboard. You know the edges of the cardboards are sharp. I’m old and my skin is fragile. I can easily bleed from broken skin.” (I15).

#### 3.2.4. Theme 4: Family Relations

Another key theme, identified across 20 older adult IWPs (143 references) and 2 key informants (2 references), relates to the family relations of the older adult IWPs. Most of the discussion regarding family relations were related to conflicts between the older adult IWPs and their adult children. The children did not support their older adult parents in picking. As illustrated by a participant who picks with his wife, “They [their children] said that I am too old for this job and this job has low reward, so there is no need for me to do it. I said, I can’t. Your mother also can’t. It’s tricky because if you stop her she said that she will get sick. She doesn’t feel comfortable.” (I03).

Many of the older adult IWPs shared how the conflict emerged from the stigma attached to picking, as a participant opined, “I don’t feel embarrassed. This is others’ feelings, not mine. But my sons said it is a low-class job and others will look down on me. Others may discriminate me and my children. I don’t rely on them so it shouldn’t bother them. They said they are somehow affected. Yes, their classmates may think they have financial problem because I scavenge cardboards. They are worried about such sayings and how it affects them.” (I10).

Another participant echoed this sentiment, 

“Well, they [her children] want me to stop but they can’t stop me. Like now, I am able to walk and move. If I don’t find something to pass the time, I feel very lonely. A lot of things to think about, sitting at home with just your four walls facing you. You die sooner.” (I15).

The conflict between the older adult IWPs and their children was also identified by one of the key informants. An academic who has done research with this group shared,

“This [referring to the children telling their parents to stop picking] increases their burden. It’s like the children say don’t go, we’ve already told you to stop waste-picking. Some homes don’t support them. So, they might face some family conflict. Their environment is already not the best and this adds to their stress. Life is tough.” (I15).

#### 3.2.5. Theme 5: Mindset and Values of the Older Adult IWPs 

Another key theme, identified across 24 older adult IWPs (189 references), relates to their mindset and values. While the older adult IWPs have diverse life experiences, there are common characteristics that describe the mindset and values of most of the participants: (1) used to suffering, (2) low self-efficacy and introversion, (3) strong need for independence and dignity. 

##### Used to Suffering

Almost all of the older adult IWPs shared vivid stories of their past that were characterized by significant adversities, including for example starvation, family separations and/or death of parent(s) at a young age, extreme poverty, World War II, and the Cultural Revolution, which have shaped their mindset and values in the present moment. As a participant illustrates, 

“Life was quite difficult back then. Yeah, my whole person was cultivated from it. If I don’t cultivate myself I would have been on the wrong path. At that time we had nothing—no social welfare. We could not even borrow money from others. Those rich relatives -they were afraid of being dragged down, so we had a difficult time. We could not count on anybody.” (I03).

Such adversities they would note are incomparable to the challenges they experience currently. As a participant shared, 

“We are used to living in poverty. We’ve come here [Hong Kong] for more than 20 years. We came in 1994 and our life was so hard. We scavenge cardboard now and can get around HK$10. It’s like getting gold! We are used to living in poverty, we were so poor.” (I04).

A survival mindset was further illustrated by this participant who responded with the following when asked what sort of activities she would like to do if she had free time, “I don’t have such thinking, such as like and dislike. Yes, everyone just needs to survive, right?” (I02). Another older adult IWP shared, “I grew up with hardship. It was so difficult you know. Right now I have no worries. I have nothing, so I will not lose anything.” (I08).

##### Low Self-Efficacy and Introversion

Some of the older adult IWP shared that a reason why they do not attend some leisure activities provided by NGOs is that they prefer to be alone. As one participant shared, 

“No, I don’t have other activities. What can I do? I don’t like gathering. I love staying alone. I am not a good person. Yes, I seldom socialize with others. I am afraid of saying something wrong. It is not good if I offend someone.” (I04).

Another participant echoed this sentiment explaining why she is so happy with her life right now, “[I am] happy [that] I live alone! I live alone and I can plan my schedule by myself. I don’t like to talk too much.” (I08). Some older adult IWPs also shared comments highlighting low self-efficacy, as illustrated by this participant “Yes, so I don’t think too much. I have no brain capacity to think too much. I am so stupid as a pig.” (I07). Another older adult IWP shared she does not go to social activities with groups because she is not educated, 

“I don’t talk too much because I don’t know how to say things, I am not educated. I don’t want to offend others, right? So I go out and be a silly woman. To scavenge something. Then the next thing you know the day is filled!” (I11).

##### Strong Need for Independence and Dignity 

While some older adult IWPs shared a sense of low self-efficacy, there were others that shared a strong need for independence and dignity. As illustrated by this participant, 

“Many people donate things to me, but I reject. I don’t want to help others and I don’t want to seek help from others. Yes, rely on myself. Others don’t seek my help and I don’t seek others’ help. It is difficult for me to return the favour. I put this in my mind.” (I07).

Another participant shared a similar experience of receiving donation or charity from others and rejecting it, 

“I don’t want to have their money. I will feel miserable if I receive their donation. They said they gave me out of concern, no other reason. I rejected and told them to keep their money. And thanked them for their kindness. Sometimes they would also donate to my wife, but I told her to reject it. I don’t need any donation to support my living condition. It is not good to receive other’s donation.” (I10).

Another participant echoed this sentiment sharing “I never seek help from others. I have never borrowed a single penny from others. I earn with my hands.” (I01). Some participants even shared how they would seldom seek help from their family, as illustrated by this participant “I seldom find him [referring to his son]. Yes, I seldom seek help from others.” (I06).

#### 3.2.6. Theme 6: Access to Social Welfare 

Another key theme identified across 22 older adult IWPs (146 references) and six key informants (12 references) relates to access to social welfare. The older adult IWPs shared various concerns and experiences that highlight challenges related to accessing social welfare, whether it be for housing, medical needs, and/or basic income to meet daily needs. As one participant shared her problems with applying for public housing, 

“The most important thing for me is to get public housing now. I am still waiting for it. I don’t know when I can get it. Maybe 3 or should be 4 years? I don’t know, they gave me a form. I have to fill in that form, but I still have not received any approval. My landlord said my contract will end very soon and she requested me to leave. Because the building is going to be demolished. I have called the landlord and asked if I can live one more month and she allowed for it. But two days before, she suddenly called me and requested me to leave. So I have to rent a new flat immediately. I have no choice.” (I02).

Another participant who did not have public housing echoed this sentiment, 

“The fruit money (OAA), it’s not enough. Things are expensive nowadays. If you have to pay rent with HK$3000 dollars you don’t even get to eat after you pay rent. You know? I applied and I think I am a year away from the three years waiting period. We will see. We old people—it’s faster for us, but I don’t know how much faster.” (I15).

Another participant shared how despite being eligible for social welfare, he refused to apply, 

“If I have the social welfare. I will no longer need to do waste picking for making up the living expenses. I don’t need to enjoy life. Just no need to suffer. I did not apply—people said to me why didn’t you apply for the social welfare? I don’t want to count on it when I can still make my own living. So I only have the fruit money [OAA]. It’s only slightly over a HK$1000.” (I03).

Another participant shared how her OALA (means-tested financial aid) was cancelled and had some confusion about whether the OAA (non-means tested financial aid) was also cancelled, 

“They cancelled my OALA. They said I had HK$100,000–200,000. So I do save money in the bank. They have my record in the bank and cancelled my OAA. Does cancel mean I can get nothing? I am afraid of getting nothing after they cancel. They sent a letter to me. I don’t know how to fill it in.” (I05).

Another older adult IWP shared how she would seldom access the public hospital due to the fees to book an appointment, “I seldom visit the doctor. You need to book an appointment. HK$50 for an appointment, but I can earn HK$50 from scavenging cardboard only.” (I07).

Another older adult IWP, who was not yet 65 years of age and had some MPF, shared her initial struggles with applying for CSSA, and that she ended up having to pay for a significant out-of-pocket medical expense. Albeit, with the help of an NGO and a social worker she was able to finally receive CSSA and also reimburse the medical expense incurred. She explained,

“Then because I got diabetic retinopathy that doctor said I needed to receive injections but I didn’t have money. I had 4 injections for this eye. That was the final injection so it was more expensive. It was HK$9700. They [the NGO] helped me to claim back the fee. They are so nice. I didn’t know them before but they helped me. Later [the social worker] usually comes and visit us. We discussed my case and I got CSSA. I found someone to help. As you know we don’t know many things.” (I09).

Some key informants also shared how access to social welfare was important for this group, as a social worker explained, “And if there is government cash-handout at the time or if there are any social welfare programs, we’d tell them about it and help them apply.” (I12). Another key informant, a former FEHD officer, opined that the limited social welfare for older adults is a problem, 

“Why must they pick to sustain their lives? Why can’t they have a comfortable life in their later life stage? You see almost all of the scavengers are older in age, but not one of a younger age. Why would there be such a poor situation in older people’s welfare? So this is the fault of the society because of the inadequacy of the older people welfare.” (I14).

#### 3.2.7. Theme 7: Formal versus Informal Work 

Another key theme identified across 13 older adult IWPs (48 references) and nine key informants (28 references) relates to formal versus informal work. The key informants had mixed perspectives regarding formalization of picking (making picking a formal occupation). A district councillor shared, 

“Well, if we turned it into a business maybe it’ll be a good thing if we can make it work. Like territorialisation and conflict between waste-pickers. Why? Because there’s no official order and so they need to create some sort of underground order. And these problems arise because you didn’t recognise them as official workers from the beginning.” (I08).

In contrast, a social worker shared the following, 

“It is about their options. The special feature of waste-picking is that it is easy to enter the industry and basically we can go and start working now ha ha ha. You do not need an employer, a contract, a departmental official, and you do not need to prepare anything to start working.” (I09).

Another key informant from an NGO shared the following, 

“Of course there will be two outcomes if there is a law regulating waste-picking as formalised work. The first outcome is waste-picking will become systematic, just like the outsourcing system of FEHD cleaning. I think it is very unlikely that the government would legislate to regulate waste-picking because it is too complicated. Once it is regulated, some parties will be eliminated, and you know who will be eliminated—the older people. I think the best is to keep the current situation, because nobody wants to get a hand on this as it is too complicated. It is better to keep this state (informal work) because nobody is going to be eliminated or regulated.” (I06).

While the older adult IWPs did not discuss formalization in particular, many of them (*n* = 12, 47 references) shared how the flexibility of their work is important, as illustrated by this participant, 

“I can quit anytime, but sitting here is like waiting for death right? I can’t do it for long hours as well. It’s just 2–3 h and then I will have a rest. It works fine. I am too old to be hired. This job is the most suitable for me. I don’t have to be rushed. I can earn a little money, something like this.” (I03).

Another older adult IWP echoed this sentiment, 

“There is no restriction of scavenging, right? I pick some when I have time. You don’t need to count the amount and there is no constriction. I feel happy about this. This is not a job, right? You can use any method to fold. No rules, and I do it when I have time. You have freedom. Jobs don’t have freedom, right?” (I04).

#### 3.2.8. Theme 8: Prices for Recyclables

Another key theme identified across 18 older adult IWPs (58 references) and 14 key informants (56 references) relates to the prices regarding recyclables. Many of the key informants shared diverse perspectives on the recent subsidies from the government that enabled the minimum price of 1 kg of cardboard to almost double from HK$0.4–0.5/kg to HK$0.7/kg. A recycling depot staff shared the following, 

“Well, with the subsidy the price for buying cardboard was fixed at HK$0.7/kg, but then we also sold the cardboard for HK$0.7/kg. The price of HK$0.7/kg helped the grannies, and we sold the cardboard for HK$0.7–0.8/kg. So the subsidy was obviously trying to help the grannies by raising what we would pay them. We didn’t earn much from selling the cardboard, but the government subsidised us. If we sold 25 tons of cardboard, then we’d get HK$9000. If we sold 40 tons a month, then we’d get HK$13,000. So we’d get money from the subsidy to cover our costs.” (I02).

In contrast, this key informant from the recycling industry echoed this sentiment, 

“It is not always good to set the buying price of paper too high, because it will harm the older people. My personal evaluation is that the best price is about HK$0.6–0.7/kg, because the younger generation around the 40 to 50 years age would not have strong incentive to collect cardboards at this price. So, waste-picking would be friendlier to the older people. If it is over HK$1/kg, there will be competition as more people would do waste-picking. You know older people cannot walk rapidly, they cannot collect cardboards as fast as others.” (I07).

A district councillor shared how the price setting does not even really matter as he explained, 

“Actually, I think there is not much actual impact. The problem is it is very difficult to monitor. Let’s say the buying price is fixed but the depots could cheat by adjusting the measurement of the weight so they can pay less to the waste-pickers. This is what I mean like when we see waste-picking as a business. The businessmen have many tricks to maximise their profit and it turns out to exploit the pickers.” (I11).

The older adult IWPs also talked about how the price of the recyclables impacted their work. One participant shared that she used to have problems finding a recycling depot to collect her recyclables when the price was set at HK$0.3/kg. She explained,

“I used to put the trolley full with cardboards on the street overnight and carry and sell it to the recycling depot in the morning after around 7–8 a.m. But you know, at that time the price of cardboards was really low, which was only HK$0.3/kg, so not many depots would buy the cardboard that I collected.” (FGD, P2).

At the same time, many of the older adult IWPs shared how there is an increase in competition that has impacted their work. As a participant shared, 

“Yes, more people started scavenging like what I’ve mentioned before, because the recovery price is high now. When I walk around now some youngsters like you they started scavenging too so I can’t get any. It was HK$0.3/half kg in the past so I got more. But I can’t get any cardboards now. That’s it [pointing to her trolley].” (I04).

#### 3.2.9. Theme 9: Impact of COVID-19 

Another key theme identified across 12 of the 31 references relates to the impacts of COVID-19. Mostly, the older adult IWPs shared that the number of recyclables has decreased since the start of the pandemic. One participant explained “Because fewer people are shopping now, so there are fewer cardboards now.” (I02). Another participant echoed this, “Yes, poorer conditions during this new wave of outbreak. Yes, no one buys food and fruits. How can we get cardboards? Yes, many shops close down. Yes, I have to check rubbish bins so many times.” (I09). 

### 3.3. Contributions (Social Impact) of the Older Adult Informal Waste Pickers

#### 3.3.1. Theme 1: Caregiving to Family Members

While not the majority, a unique theme was identified in the older adult IWPs interviews (*n* = 7, 39 references), which relates to the older adult IWPs as caregivers to their family members. One participant shared that because of her caregiving duties to her grandchild she has to continue to pick, 

“I need to take care of my grandchild. I have this difficulty only. My grandchild has this problem. I have no solution. My son’s health is not good. He has a disease. He needs to work. There is no solution, so I scavenge cardboards.” (I07).

Another participant shares how her son’s mental health is a constant worry for her, 

“My son has anxiety. So, I have this worry now. I am worried about the recurrence of his problem. He is now taking medication. There will be no harm to his health if he can still take medication! But he doesn’t listen to me. He is so aggressive. I am worried about him now.” (I04).

#### 3.3.2. Theme 2: “Environmental Pioneers” and Enhancing Public Cleanliness 

Almost all the key informants (*n* = 13, 32 references) shared that the older adult IWPs provided a significant contribution to the community and society, specifically as “environmental pioneers” and by enhancing public cleanliness. This is illustrated by a staff form the recycling depot, who explained “but I tell them they don’t have to be embarrassed, that they should think of what they’re doing as doing something for the Earth as something environmental.” (I02). A shop owner echoed this sentiment, “waste pickers are actually doing a job benefiting the environment. They have their contributions. Imagine without waste-pickers, no one would do such work.” (I04). Another key informant, social worker, opined, 

“From the perspective of the recycling industrial chain, waste-pickers are the frontline workers. Let’s suppose waste-picking and recycling is a job. They have long working hours of waste-picking and are the bottom of the pyramid of the recycling industry. They are the first one to interact with the recyclables. So, I would say they are the environmental pioneers. I want them to gain that recognition.” (I06).

The key informants were also asked in the interviews to describe the older adult IWPs in one or a few words. These responses were analysed using word cloud. “Environmental pioneer/protection” was the most frequent word used to describe the older adult IWPs from the key informants’ perspectives. The word cloud is illustrated in Figure 1.

Discussions about social contribution of their work were identified across 23 older adult IWPs (93 references). Unlike the key informants, the older adult IWPs’ perspectives on their social contributions as environmentally friendly and enhancing public cleanliness were mixed. About half who believed they made a social contribution, especially to public cleanliness. As one participant shared, 

“I take this as helping them to throw away the garbage. It’s mutual help. I take my heart out to help people. I am getting older. I want to accumulate merits. I clear the waste, and the outsourced workers (referring the formal cleaning staff in public areas) are less burdened. If I don’t do that, the waste would be all over the street, a total mess.” (I03).

Another participant shared, 

“Well, we both get something. To society the cardboards are taken to be recycled. More income to them. And to us too personally. If we don’t do that, the government would have to hire a lot more cleaners.” (I12).

In the interviews the older adult IWPs were asked to complete the sentence “without the older adult IWPs, Hong Kong would be…” The responses were analysed using word cloud. The word cloud also supports this theme that the older adult IWPs contribute to public cleanliness, with most frequent words as “cardboards everywhere” and “dirty streets.” The word cloud is illustrated in Figure 2. 

At the same time, the other half of the older adult IWPs thought their work provided no social contributions. One participant said, “How can you provide social contributions from this. I don’t think it is a kind of social contributions. No, you do this because of money. You want to earn money so you scavenge. Earn money only. Not for other reasons. It is the truth”. (I04). Another participant echoed this sentiment “Environmentally friendly? What? No, no relationship. Nothing! Just rubbish!” (I08). 

## 4. Discussion

The themes inform recommendations that cover a range of individual, family, and community service responses to address healthy ageing of this unique sub-group. First, is the prioritization of the mindset and values of the older adult IWPs when developing and implementing healthy ageing initiatives. The older adult IWPs are a diverse group with different personalities. At the same time, there are still common experiences (e.g., war, extreme poverty) that have shaped their mindset and values influencing their motivations, behaviours, and decision-making. This theme suggests that policymakers and practitioners should seek to explore, understand, and prioritize the mindset and values of the older adult IWPs when designing and evaluating policies and services to support their well-being. Recommendations include, for example, starting and/or continuing to: (i) acknowledge the importance of perceived/subjective well-being as outcome indicators (alongside with objective measures), (ii) using a “first-mile” (rather than “last mile”) approach to service responses, which means integrating the service users’ (older adult IWPs) mindset and values at the very beginning of service design, (iii) prioritizing values of autonomy, agency, independence, and dignity in initiatives, and (iv) recognizing generational differences (e.g., between practitioners and the older adult IWPs) in how outcomes, such as “well-being”, “happiness”, “purpose/meaning”, and “health” are defined.

Second, is a continued focus on diversity in designing and developing healthy ageing initiatives with a particular focus on understanding the needs of unique sub-groups that lie on the margins of society. As the older adult IWPs age and experience age-related decline (theme 3) they may need to do less picking or even stop picking altogether. Alternative activities to picking are needed to help transition and prepare the older adult IWP to continue participating in social participation activities, which are a part of healthy ageing [12]. The theme highlights that even though older adult IWPs are invited to participate in different activities, they often choose not to, because they prefer to pick. Thus, the issue is not the limited number of activities available but maybe the types of activities. Recommendations for practice, include for example starting and/or continuing to offer alternative activities that consider the trifecta experience, whereby the activity can address all three preferences of the pickers: provide some income, provide exercise, and be meaningful/purposeful to the older adult IWP (e.g., tangible tasks are involved, sense of tangible accomplishments/completion can be experienced, is work/employment-like activity). 

Third, is the recommendation that healthy initiatives aim to facilitate a strong sense of self-efficacy among older adults. The findings (theme 5) indicate that many of the older adult IWPs have low self-efficacy. People with low self-efficacy have little faith in their own capacities, abilities, and power to create change within their own lives. Low levels of self-efficacy have been linked with negative mental health outcomes (e.g., stress and depression) (Choi et al., 2020) [13]. In contrast, people with a strong sense of self-efficacy believe that they have the skillset or at least the capacity to learn the skillset to address challenges and problems that come their way. In other words, they believe they are in control of their own lives and have the power to create change. High levels of self-efficacy have been associated with higher quality of life and adoption of health-promoting behaviours (Shaabani et al., 2017) [14]. Therefore, a recommendation for service response is to facilitate a strong sense of self-efficacy of the older adult IWPs. Efforts toward this aim may include for example healthy ageing initiatives that: (i) address the internalized stigma (that is, transforming the older adult IWPs’ own negative perceptions of picking to an activity with a social impact of community recycling), (ii) focus on identifying existing strengths and skillsets of older adult IWPs and creating opportunities for them to contribute to society through such capacities, and (iii) build new skillsets and strengths of older adult IWPs and creating opportunities for them to contribute to society through the newly acquired capacities. 

Fourth, is to see healthy ageing initiatives through a family lens. The older adult IWPs shared their concerns regarding their family issues. This theme suggests that family relations play a role in influencing their wellbeing. Recommendations for practice include for example starting and/or continuing to adopt a “family-systems” approach [15] to support the older adult IWPs. Noting the diverse family issues experienced by the pickers and that support for the older adult IWPs’ well-being may also include understanding and supporting the needs and well-being of their family. Social workers can play a particularly impactful role in addressing this service need, as one of the professional roles of social work is to help family members to enhance relationships with each other and to cope with adverse situations. Lastly, this theme supports public awareness campaigns to destigmatize the shame of picking (same as theme 1), as this seems to contribute to family conflicts between the older adult IWPs and their adult children. 

Fifth, is facilitating picker-friendly neighbourhoods. A key finding (theme 1) highlights that the environment such as the neighbourhood plays an influential role in the healthy ageing experiences of the older adult IWPs. Thus, a recommendation for future research and practice is to explore and facilitate picker-friendly neighbourhoods. Working towards picker-friendly neighbourhoods include two dimensions: social and physical, with an emphasis on the former. Efforts towards enhancing the social dimension of picker-friendly neighbourhoods include for example: (i) service activities that build better connections and relations between older adult IWPs and the recycling industry and FEHD (e.g., encouraging a friendly and respectful attitude towards the older adult IWPs), and among the older adult IWPs themselves; (ii) service activities that destigmatize picking (e.g., changing the community’s perceptions of picking to see it as a valuable and contributing activity that has a social impact of community recycling; encouraging the community to replace derogatory terms such as “scavengers” with “community-recycling workers” or “community-recycling pickers”). Efforts towards enhancing the physical dimension of picker-friendly neighbourhoods include, for example, service activities that enhance the working environment and conditions of picking. For example, service providers can help to advocate with the FEHD and the recycling industry for a safe and secure physical workspace in the community (e.g., storage lockers available for older adult IWPs to safely store their cardboard overnight so that they do not have to resort to sleeping on the streets to safeguard their supply).

Finally, healthy ageing initiatives for this group of older adults also require a macro/policy lens. Theme 6 highlights that public housing and financial social welfare play a key role in the well-being of the older adult IWPs. Those who are not accessing such assistance may be the most vulnerable and require support in navigating the welfare system and accessing such supports. These include, for example, those who are the youngest and have some savings (but not a significant amount to sustain past 1 to 2 years of basic living) and thus are not eligible for public housing and/or other types of government financial aid; those who migrated from the Mainland more recently (e.g., less than 7 years) and/or have not continuously lived in Hong Kong to be eligible for the OAA; those who are on the waiting list for public housing and those who refuse to apply despite their eligibility. 

Practice recommendations include, for example, starting and/or continuing to adopt a case management approach to addressing the diverse social welfare needs and situations of the older adult IWPs, especially with a focus on the unusual/unique cases, whereby the older adult IWPs do not easily fit into the criteria stipulated by such policies but in fact need such social welfare. Services responses should not only help the older adults navigate and access social welfare, but also focus efforts to destigmatize accessing social welfare (e.g., seeing it as a right rather than a pity/charity situation). Lastly, a common situation is the older adult IWPs who have chronic or acute health issues that require expensive out-of-pocket medical attention and care. Focusing on supporting this group through connecting them to various medical-focused social welfare programs like the community care fund medical assistance program in Hong Kong, is another recommendation. 

## 5. Conclusions

Healthy ageing outcomes are influenced by a complex and dynamic interplay between the individual and the environment in which they live (Clark and Nieuwenhuijsen, 2009) [16]. Healthy ageing initiatives need to focus on the individual and on creating environments for healthy ageing. Creating a socially inclusive environment is a critical component of healthy ageing. Yet, it is a dimension that is often neglected. A comprehensive and multi-level approach is needed to foster the healthy ageing needs of older adult IWPs and these recommendations recognize a need to focus on both the individual and their social environment.

The study sample was specific to a unique sub-group of older people in Hong Kong, the older adult IWPs or, locally called, the “Cardboard Grannies”. Thus, a limitation of this study is the generalizability of the findings. In particular, the study cannot provide statistical generalizability, but can facilitate analytical generalizability and transferability [17,18]. The write-up of the findings is descriptively rich to provide the reader with a vicarious experience and in-depth details so that they can make an informed decision regarding the transferability and utility of the findings to their own context and service-user populations. Indeed, some of the issues and recommendations identified may resonate with other marginalized groups of older adults. An in-depth examination of the lived experiences of older adults who lie on the margins of society are needed to ensure that no one is left behind in healthy ageing.

## Figures and Tables

**Figure 1 ijerph-19-09691-f001:**
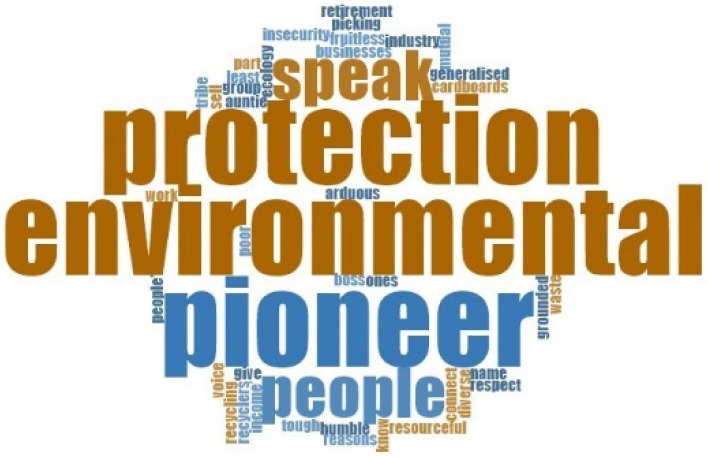
Key Informants’ descriptions of the Older Adult IWPs.

**Figure 2 ijerph-19-09691-f002:**
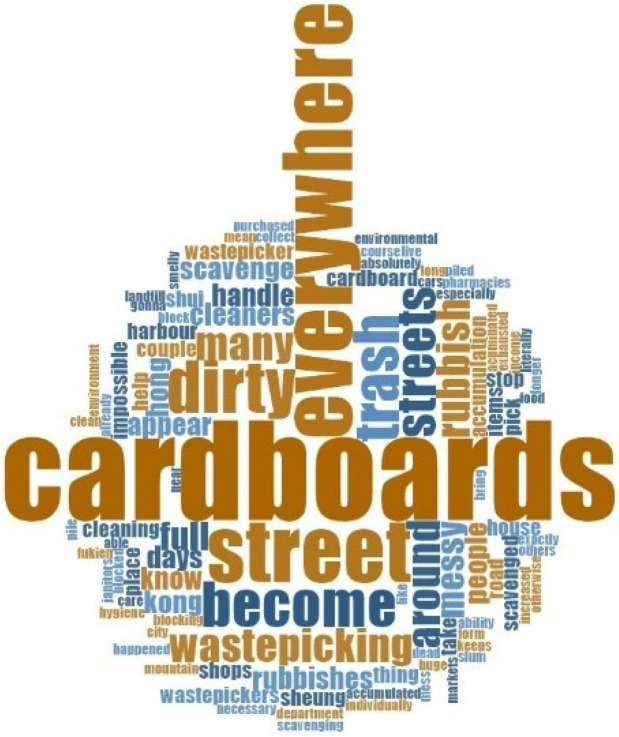
“Without the Older Adult IWPs, Hong Kong would be…” (older adult IWPs’ responses).

## Data Availability

Data supporting reported results can be given upon request of the first author.

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
