# Peer review of "Leaving No One behind in Healthy Ageing: A Unique Sub-Group, the “Cardboard Grannies of Hong Kong”"

_ijerph, 2022, doi:10.3390/ijerph19159691_

Round 1
Reviewer 1 Report
Thank you for the opportunity to review the manuscript.
In my opinion, this is a meaningful study, which has positive significance for promoting Healthy Aging of the special sub-group "Cardboard Grannies of Hong Kong". I don't have any major comment, but only a few minor ones are shown below.
1) Line 83 shows that the sample number of older adult IWPs is 25, and line 110 shows that the total number of Participant Characteristics is 40. However, lines 111, line 128 and Appendix A show that the sample number of older adult IWPs is 26. Please carefully check whether the data are correct when revising, so as to avoid this kind of data inconsistency.
2) The expression of the interview content in the section 3.2 and 3.3 (line 136~759) are a bit messy, and the use of punctuation marks is also very casual, which greatly weakens the readability of this article and reduces the readers' favorable feeling. Before submitting the revised version, the authors should strengthen the standardization and orderliness of these sections, and adopt some marks and symbols to improve the readability of the interview content, rather than just listing what the interviewee said.
Overall, this is a really interesting, well-designed and well-researched paper. I really enjoyed reading it.
Author Response
Thank you very much for taking the time to review the manuscript and providing feedback.
Response to Reviewer 1:
Response to Reviewer 1:
1) Line 83 shows that the sample number of older adult IWPs is 25, and line 110 shows that the total number of Participant Characteristics is 40. However, lines 111, line 128 and Appendix A show that the sample number of older adult IWPs is 26. Please carefully check whether the data are correct when revising, so as to avoid this kind of data inconsistency.
The final sample of older adults is n = 26 (making the final sample N =41) and has been corrected on line 83 and 110, respectively.
2) The expression of the interview content in the section 3.2 and 3.3 (line 136~759) are a bit messy, and the use of punctuation marks is also very casual, which greatly weakens the readability of this article and reduces the readers' favorable feeling. Before submitting the revised version, the authors should strengthen the standardization and orderliness of these sections, and adopt some marks and symbols to improve the readability of the interview content, rather than just listing what the interviewee said.
In section 3.2 and 3.3. further subheadings/numbers were created to indicate the separate subthemes. It is now the following:
3.2. Service Needs: Perspectives from Older Adult IWPs and Key informants
3.2.1. Theme 1: Social Relations with Others at the Worksite/Neighborhood
3.2.1.1 Positive Experiences
3.2.1.2 Negative Experiences
3.2.1.2.1 Insecure working spaces
3.2.1.2.2 Conflict with FEHD and Other Informal Waste Pickers.
3.2.1.2.3 Stigma Attached to Picking
3.2.2. Theme 2: Picking Provides a Trifecta Experience
3.2.2.1 For Meaning/Purpose
3.2.2.2 For Income
3.2.2.3 For Exercise
3.2.2.4 Trifecta Experience
3.2.3. Theme 3: Physical Decline and Medical Health Needs
3.2.4. Theme 4: Family Relations
3.2.5. Theme 5: Mindset and Values of the Older Adult IWPs
3.2.6. Theme 6: Access to Social Welfare
3.2.7. Theme 7: Formal versus Informal Work
3.2.8. Theme 8: Prices for Recyclables
3.2.9. Theme 9: Impact of COVID-19
3.3. Contributions (Social Impact) of the Older Adult Informal Waste Pickers
3.3.1. Theme 1: Caregiving to Family Members
3.3.2 Theme 2: “Environmental Pioneers” and Enhancing Public Cleanliness
Further, we have reviewed the entirety of the manuscript carefully (sentence by sentence) to make further corrections to ensure that it attunes to language conventions and addresses grammatical issues. Especially recognizing that the comma (,) was used too often. Please see the tracked changes revisions throughout the entire manuscript noting these grammatical changes.
Reviewer 2 Report
Thank you for the opportunity to review this interesting paper. My comments are more stylistic than substantive. Your use of commas throughout needs to be reexamined. Also, there are several incomplete sentences (Lines 41-42). There are other incomplete sentences that I believe are meant to be subheadings. For a non-Chinese audience, you need to explain what a "cattie" means (Line 342). There are also some errors that are not picked up by spell-checking, such as "form" where you mean "from" (Line 725). There should be a consistent use of quotation marks around actual quotes. You vary from using quotation marks within a paragraph to not using then when the quote is its own paragraph. This approach is very difficult to follow.
In the Discussion section, you do not address possible changes to the FEHD or other formal organizations that provide services to this population. Yet from your data, it would seem that leaders within these agencies could be helpful in improving the situational and environmental concerns of these older adults. For example, having to sleep on the street to watch over their cardboard makes them vulnerable to violence, disease, and exposure. Having storage lockers available for these individuals where they could safely store the cardboard overnight would help in this regard.
Author Response
Thank you very much for taking the time to review the manuscript and provide feedback.
Response to Reviewer 2:
Thank you for the opportunity to review this interesting paper. My comments are more stylistic than substantive. Your use of commas throughout needs to be reexamined. Also, there are several incomplete sentences (Lines 41-42). There are other incomplete sentences that I believe are meant to be subheadings. There are also some errors that are not picked up by spell-checking, such as "form" where you mean "from" (Line 725).
As mentioned in the response above, we have reviewed the entirety of the manuscript to make the changes to address the grammatical errors (including line 725) and the style, including the subheadings so they don’t look like incomplete sentences anymore (Lines 41-42).
For a non-Chinese audience, you need to explain what a "cattie" means (Line 342).
The following was added to explain what a “cattie” means:
“I can buy a few catties [a traditional unit used in Hong Kong equivalent to about 600 grams] of vegetables. But I can’t buy one catty of fish! I can’t buy one catty of pork too. (HK)$68 for one catty of pork. I can’t afford that, right?” (I07).
There should be a consistent use of quotation marks around actual quotes. You vary from using quotation marks within a paragraph to not using then when the quote is its own paragraph. This approach is very difficult to follow.
All quotations were revised in the manuscript to be consistent. If the quote was 40 words or more than it was made into a block quote (indented) and quotations were added to make it easier for the reader to know it is a quote. If the quote was less than 40 words it was included in the paragraph, and quotations remained to show to the reader it is a quote.
In the Discussion section, you do not address possible changes to the FEHD or other formal organizations that provide services to this population. Yet from your data, it would seem that leaders within these agencies could be helpful in improving the situational and environmental concerns of these older adults. For example, having to sleep on the street to watch over their cardboard makes them vulnerable to violence, disease, and exposure. Having storage lockers available for these individuals where they could safely store the cardboard overnight would help in this regard.
The following (underlined text and highlighted) was added to the fifth discussion point on p. 18:
Efforts towards enhancing the social dimension of picker-friendly neighbourhoods include for example: (i) service activities that build better connections and relations between older adult IWPs and the recycling industry and FEHD (e.g., encouraging a friendly and respectful attitude towards the older adult IWPs),…
Efforts towards enhancing the physical dimension of picker-friendly neighbourhoods include for example: (i) service activities that enhance the working environment and conditions of picking. For example, service providers can help to advocate with the FEHD and the recycling industry for (e.g., advocating for a safe and secure physical workspace in the community (e.g., storage lockers available for older adult IWPs to safely store their cardboard overnight so that they do not have to resort to sleeping on the streets to safeguard their supply).